# Health education improves referral compliance of persons with probable Diabetic Retinopathy: A randomized controlled trial

Zara Khair[1][*], Md Moshiur Rahman[1], Kana Kazawa[1][‡], Yasmin Jahan[1][‡], Abu S. G. Faruque[2], Mohammod Jobayer Chisti[2], Michiko Moriyama[1]

1 Graduate School of Biomedical and Health Sciences, Hiroshima University, Hiroshima, Japan, 2 Nutrition and Clinical Services Division, International Centre for Diarrhoeal Disease Research, Bangladesh (icddr, b), Dhaka, Bangladesh

☯ These authors contributed equally to this work.
‡ These authors also contributed equally to this work.
* zara.oikee@gmail.com

**Data Availability Statement:** Data are available in figshare: https://doi.org/10.6084/m9.figshare.13172837.v1 (10.6084/m9.figshare.13172837).

## Abstract

### Objective

Lack of awareness about Diabetic Retinopathy (DR) is the most commonly cited reason why many persons with type 2 diabetes are non-compliant with referral instruction to undergo retinal screening. The purpose of this study was to evaluate the efficacy of a culturally, geographically and socially appropriate, locally adapted five-month-long health education on referral compliance of participants.

### Method

A prospective randomized, open-label parallel group study was conducted on persons with type 2 diabetes who underwent basic eye screening at a diabetes hospital between September 2017 and August 2018. Participants who were noncompliant with referral instruction to visit a hospital for advanced DR management were randomly divided into health education intervention group (n = 143) and control group (n = 156). Both groups received information regarding DR and referral instruction at the diabetes hospital. The intervention group was provided personalized education followed by telephonic reminders. The primary endpoint was 'increase in referral compliance' and the secondary endpoint was 'increase in knowledge of DR'. Multivariate logistic regression model was used to identify significant predictors of compliance to referral.

### Results

A total of nine participants dropped and 290 completed the post intervention survey. The compliance rate in intervention group was found to be significantly higher than the control group (64.3% vs 28.2%; OR 4.73; 95% CI 2.87–7.79; p<0.001). Participants in the intervention group acquired better knowledge on DR (p<0.05). Apart from intervention, referral compliance rate was also found to be significantly associated with participants' self-perception

**Funding:** The author(s) received no specific funding for this work.

**Competing interests:** The authors have declared that no competing interests exist.

of vision problem (OR 2.02; 95% CI 1.02–4.01; p = 0.045) and participants' income (OR 1.24; 95% CI 1.06–1.44; p = 0.008).

## Discussion

Our results suggest that intensive health education on DR should be integrated with diabetes education as it may result in significantly improved referral compliance. Outcomes may be sustainable if interventions are institutionalized at referral point.

## Trial registration

Clinical Trials.gov (Registration # NCT03658980); https://clinicaltrials.gov/ct2/show/NCT03658980.

## Introduction

Type 2 Diabetes Mellitus (T2DM) is on an alarming rise in the world [1]. In 2019, approximately 463 million adults (20–79 years) were living with diabetes and by 2045 this will rise to 700 million [2]. Bangladesh has been identified as one of the top ten countries worldwide in terms of the number of people living with diabetes [3]. In 2019, there was an estimated 8.4 million people with diabetes in the age bracket 20–79 in Bangladesh, and this number is predicted to increase to 15 million by 2045 [4].

Diabetic Retinopathy (DR) is the leading cause of vision loss in adults aged 20–74 years, and the fifth leading cause of global blindness [5,6]. In Bangladesh, the national prevalence of diabetes is estimated to be 8.4% in population aged 20–79 years, with approximately 56% of them undiagnosed [4]. The estimated number of individuals with DR in Bangladesh is 1.8 million (21.6% of people with diabetes) [7]. Higher prevalence of DR has been reported in the coastal areas compared to rural population from other areas of Bangladesh, according to a study that was conducted in Barishal Division [8].

Since DR is initially asymptomatic, many people with diabetes are not aware that their eye condition, if left unmanaged, may affect their vision and lead to blindness [9]. Screening and early intervention is critical—it is more cost effective and may result in better health outcomes [10,11].

In Bangladesh, DR management is not yet integrated as part of mainstream public health systems and as such most registered persons with diabetes do not get their eyes routinely examined [12]. DR management services are largely unavailable in facilities designated for management of cases of diabetes, where other disease management is prioritized given limited resources [12]. This may be because of low demand for DR management services possibly due to lack of awareness about the disease, even among registered persons having diabetes. DR management services can be very expensive [13]; therefore diabetes hospitals may not cater to this service in limited-resource settings where demand for services is low.

A large proportion of persons with diabetes are non-compliant with referral to an Ophthalmologist [14] because of lack of awareness about eye complications of diabetes and lack of information regarding where services are available [15,16]. Published systematic reviews assessing RCTs found that providing health education regarding DR among persons with diabetes is a promising intervention that resulted in increased DR screening rate [17,18].

There does not exist any published RCT on said topic that has been conducted in a least developed country (LDC) or in a low and middle-income country (LMIC), although RCT is

considered the gold standard for evaluating effectiveness of health education interventions [19]. To the best of our knowledge, this is the first ever RCT conducted on said topic in an LDC or an LMIC.

Health education interventions must be contextualized according to geographical, cultural and socioeconomic needs. For instance, home-based mail reminders may not be feasible in remote rural locations in Bangladesh given low literacy rates and because effectiveness of mail reminders has been found to be 'quite modest and short-lived' in previously published studies [20]. Printed education messages alone have at times failed to increase retinal screening among persons with diabetes [11]. Interventions in limited-resource settings, where literacy rates are low, perhaps require more personalized face-to-face connections and must be trialed to study its effectiveness and scalability.

To address the challenge of suboptimal referral compliance, this study used an innovative approach that comprised of comprehensive and multicomponent modalities such as interactive face-to-face education session using demonstrative flipchart, colorful pictorial leaflet and special referral card, and telephonic follow-ups. The purpose of this study was to evaluate the efficacy of a culturally, geographically and socially appropriate, locally adapted five-month-long health education intervention on participants' referral compliance. In our research we were able to successfully evaluate the efficacy of the intervention and recommend behavior change strategies as well.

## Methods

### Trial design

This was a prospective open-label parallel randomized controlled trial designed for non-compliant participants from December 2018 to May 2019. This study was registered with Clinical Trials.gov (Registration # NCT03658980) and approved by Bangladesh Medical Research Council (Registration # 12512062018). Additionally, the study was undertaken in accordance with the Declaration of Helsinki. All participants had been explained in detail about the purpose, risks and potential benefits of the research prior to voluntary consenting and recruitment into the study. Participation was completely voluntary, and a written informed consent was obtained from all participants (S1 and S2 Files).

### Study sites

Barishal district under Barishal division of Bangladesh was selected as the study site for this study. This was done to explore the existing referral modality between a private diabetes hospital and a public tertiary hospital. This divisional level public tertiary hospital was the first in Bangladesh (outside of the capital Dhaka) to offer advanced DR management services (including eye screening using High Resolution Fundus camera, provision of treatment such as injections and laser surgeries) to the public at a very minimal cost.

### Study population

The study population consisted of participants with T2DM, registered with a private diabetes hospital and referred for advanced DR management to a public tertiary hospital.

Participants eligible for inclusion into study were adults (18 years or above) with T2DM registered with a diabetes hospital, who had undergone preliminary screening for DR using low-resolution fundus camera at the diabetes hospital between September 2017 and August 2018, were referred to a public tertiary level hospital for advanced DR management, did not

undergo a Dilated Fundus Examination (DFE) in previous 12 months and had provided informed written consent to be included in this study.

Participants who were excluded from this study were below the age of 18, persons with T2DM registered with a diabetes hospital but not referred to the public tertiary level hospital for advanced DR management between September 2017 and August 2018, persons who had undergone a DFE in previous 12 months and those who did not provide informed written consent to be included in this study.

## Statistical basis of sample size

To calculate the sample size for a statistical superiority design suitable for RCT, the formula $\{[(p_1q_1 + p_2q_2)/(p_2-p_1)^2] \times$ factor for alpha and beta$\}$ was used, where $p_1$ is the percentage of the existing referral compliance, $q_1$ is $(1-p_1)$, $p_2$ is the percentage of the expected referral compliance from intervention, $q_2$ is $(1-p_2)$, alpha is type 1 error and beta is type 2 error, and factor for alpha and beta with 90% power is 10.5.

It was assumed that the health education intervention would result in referral improvement from 35% to 55% (i.e. 20 percentage point improvement. The baseline referral rate of 35% (within control group) was assumed based on a previous published RCT conducted on persons with T2DM who were provided with personalized follow-up with the aim to increase screening for DR [21]. In this study, which involved mailing a brochure and videotape followed by telephonic reminder a week later, the referral was 31% higher in the intervention group.

During sample size calculation, a more conservative 20% increase in referral rate improvement was assumed keeping in mind the health education intervention modality (home-based health education session followed by telephonic reminders) and challenging sociodemographic and geographical context (lower education level compared to participants of aforementioned study [21], and typical barriers of accessing health care in Barishal which include lack of awareness about disease, challenging road communication, distance to facility, cost, lack of accompanying person, household commitments of women particularly living in rural areas, among others).

To detect a difference of 20% in the referral compliance using 2-tailed test with 90% power and type 1 error 0.05, the sample would be 125 in each group. The dropout rate was assumed to be 20% (more than standard 10%) because participants had already gone back home months ago following referral, and it was thought to be difficult to convince them to make the journey (possibly with an accompanying person) to the tertiary care hospital given the access barriers in Barishal and costs involved in undertaking the journey with multiple transport modalities. The total sample size was calculated to be 300 for the study.

## Randomization

To maintain research quality and reduce investigators' bias, randomization was conducted by a third person (an experienced researcher who was not related to this study). This person generated the random allocation sequence and distributed these in serially numbered sealed opaque envelopes. Envelopes were provided with either of two interventions, 'Standard Care' or 'Health Education Intervention'. The sealed envelopes were directly couriered to the tertiary hospital and were maintained in a locked cabinet under the supervision of an ophthalmic personnel. Twenty sealed envelopes were provided to field research team at a time, which was carried to participants' home and opened sequentially once a field enumerator completed the baseline in-depth interview. Envelopes were opened in a sequential manner (serial number in participant list and serial number mentioned on top of envelope was the same in all cases) in

front of study participant and a witness (usually family member of the participant), and intervention allocation was implemented accordingly.

## Outcomes

**Primary endpoint.**   The primary endpoint was 'increase in referral compliance', i.e. increase in referral completion. Referral compliance was defined as participants' visit to the tertiary hospital during a six-month period (five months intervention period and one month additional window period).

**Secondary endpoint.**   The secondary endpoint was 'increase in knowledge about DR'. Increase in knowledge about DR was measured using pre and post intervention questionnaires (S3 and S4 Files) based on a previously published study [22]. A total of nine questions were investigated to assess participants' knowledge on DR. Knowledge questionnaire was administered by experienced Community Health Workers (CHWs) at participants' homes. Study participants were asked the same questions during post-intervention survey.

## Health education intervention package for the intervention group

The duration of the total health education package was 5 months and included one face-to-face session and telephonic reminders. In case of participant being randomized to intervention group, health education session was delivered by the CHWs who completed tertiary level education and had more than 5 years' experience in disseminating eye health awareness in communities. The CHWs received orientation to deliver the intervention from the Principle Investigator (first author) who is also a doctoral candidate of health sciences major. Interventions were delivered under the supervision of Principle Investigator. The health education contents are outlined in S5 File. The contents were examined and approved by Ophthalmologists prior to field study. Telephonic reminders were delivered by the CHW who delivered the face-to-face intervention, so that participants could recognize and trust the caller as a result of rapport built during past interaction.

The face-to-face health education session was delivered in local (Bengali) language in about 30–40 minutes following an in-depth interview. It consisted of key information about Diabetes, DR, DR management options, and information about available services at the government tertiary hospital. Communication materials used were a pictorial colorful portable flipchart used for demonstration purpose, colorful pictorial leaflet in native language (Bengali) and a waterproof referral card with important information which would help to measure primary endpoint (referral compliance rate) accurately.

Telephone reminders followed this on Day 7, 30 and 90 where each reminder call lasted about 15 minutes. Multiple reminders have previously been reported to be more effective than single reminder in improving DFE [23]. A similar past study had delivered face-to-face health education session followed by monthly reminders [24]. Our study restricted telephonic reminders to three because evidence suggests that after a third patient reminder, there is no incremental improvement in screening rates [23]. Telephonic reminders were discontinued for participants who completed referral compliance prior to the next scheduled call. The contents of the face-to-face education session and telephonic sessions have been outlined in detail (S5 File).

## Control group

Participants from the control group also received information regarding DR and referral instruction from the eye health service providers at the diabetes hospital, as same as the intervention group. Participants of control group were not provided with any form of personalized

health education (face-to-face home-based education session followed by telephonic reminders).

## Statistical methods

Analysis was conducted by Intention-to-treat (ITT) method. We considered dropout participants to be those who did not receive our phone call or respond during the follow-up calls. All participants including those who dropped-out were included in analysis.

Pearsons chi-square test was used to compare the referral compliance and changes in knowledge measures of the two groups. Univariate analysis was conducted to test association of different variables with referral compliance as the primary endpoint. Subsequently, multivariate analysis with backward Likelihood Ratio (LR) binary logistic regression modeling was performed to identify significant predictors of referral compliance after adjusting for potential confounders. The significance level of association was set at $p < 0.05$ for the univariate analysis and regression analysis. The strength of association was evaluated by calculating odds ratio (OR) and their 95% confidence intervals (CI).

For the secondary outcome, we compared the proportion of participants who provided correct (or positive) responses during pre and post intervention surveys. This comparison was done between the two study arms. In addition, chi-square test was performed for each knowledge related question. There were no participants who initially provided correct (or positive) response at baseline and later incorrect (or negative) response during the post intervention survey. Therefore, we conducted chi-square test only on those participants who initially provided incorrect (or negative) responses. The significance level of association was set at $p < 0.05$ for analysis of the secondary outcome.

Statistical Software for Social Science (SPSS) Version 21.0 was used for data analysis (IBM Corp, 2012) [25].

## Reliability and validity

Due to the nature of the study design (open label), it was not possible to withhold knowledge about allocated intervention to participants. Therefore, to reduce bias, a data manager stationed at the tertiary hospital assessed outcome (compliance in both intervention and control groups). In addition, the accuracy of the outcome assessment was checked every 2 weeks by data manager and then by ophthalmic personnel at the tertiary hospital. This record was then shared with Principle Investigator.

## Results

### Baseline characteristics of the participants

The CONSORT flowchart of this study is detailed in Fig 1. Three hundred and ninety-seven persons who were referred from the diabetes hospital to tertiary hospital and met inclusion criteria were checked for eligibility. Among them, 98 persons dropped out due to reasons shown in Fig 1, and the remaining 299 persons were registered and randomly allocated to the intervention group (health education) (N = 143) or the control group (standard care) (N = 156). During study period, four and five persons dropped out from intervention and control groups respectively as we were unable to follow-up. Therefore, 139 and 151 participants were analyzed, and completion rates were 97.2% and 96.8% in intervention and control groups respectively. The socio demographic and other characteristics of study participants, obtained during baseline survey, are shown in Table 1. There were no statistically significant differences in the baseline characteristics between the two groups.

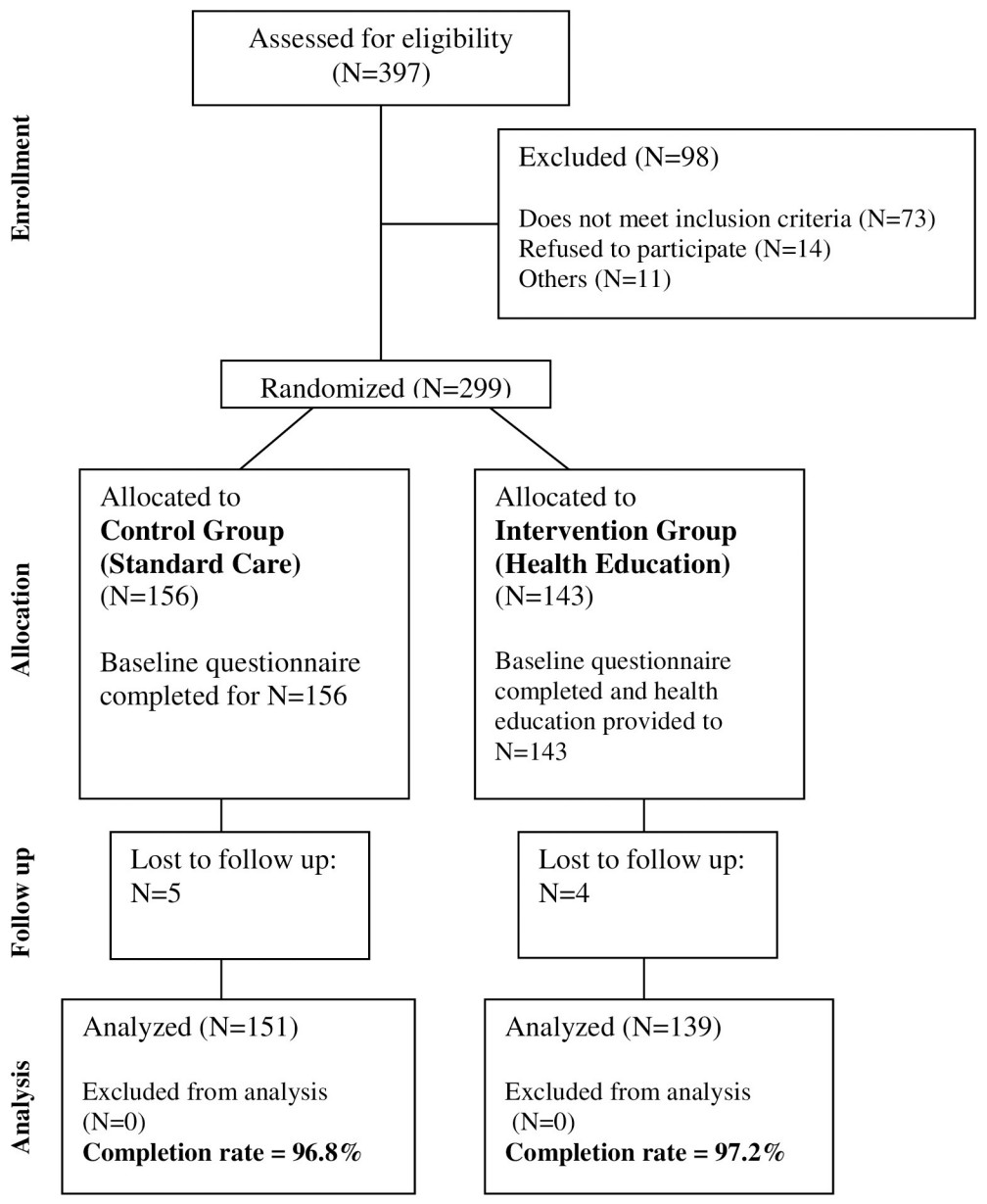

**Fig 1. CONSORT flowchart.**

### Efficacy of health education

Health education was found to be a statistically significant determinant of compliance rate (Table 2). When compared between two groups, it was observed that 64.3% participants who attended the health education session visited the referred facility, compared to 28.2% participants from control group. The difference of referral compliance rate between these two groups was 36.1% ($p<0.001$), which was higher than the anticipated 20% (OR 4.73; 95% CI 2.87–7.79).

The number of participants in each intervention group who responded positively to knowledge measures (secondary outcome) during pre and post intervention has been reported in Table 3. The post-intervention difference in knowledge levels between both groups was also

**Table 1. Baseline differences between intervention and control group.**

| Variables | Intervention Group N (%)(N = 143) | Control Group N (%) (N = 156) |
|---|---|---|
| **Demographic** | | |
| Mean age in years ± SD | 51.5±11.4 | 51.0±10.8 |
| Women | 72 (50.3) | 82 (52.6) |
| Education ≤ Grade 10 | 102 (71.3) | 110 (70.5) |
| Income ≤ BDT 9999 | 87 (60.8) | 99 (63.5) |
| **Diabetes** | | |
| Mean years of registration with diabetes hospital ± SD | 4.8±5.6 | 5.4±6.5 |
| **Eye Care** | | |
| Vision problem (self-perceived) | 120 (83.9) | 126 (80.8) |
| Duration of self-perceived eye problem < 5 years | 105 (73.4) | 105 (67.3) |
| **Referral** | | |
| Clear about referral instruction | 135 (94.4) | 150 (96.2) |

SD, Standard Deviation.

found to be statistically significant as reported in Table 4. Participants who belonged to the intervention arm and who responded negatively during pre-intervention (n = 127), were significantly more likely to respond positively to the same knowledge measures during post-intervention, when compared to the control group (n = 125).

Univariate analyses were conducted to identify sociodemographic and other factors, obtained during baseline survey, which may have had important relationship with the primary endpoint. The dependent variable was referral compliance (yes/no) including all registered participants (n = 299). The independent variables or possible predictors of compliance are shown in Table 5. Results show that in addition to the health education intervention itself, 'education' and 'income' were found to be statistically significant determining factors of participant's referral compliance. No other independent variables were found to be significantly associated with the primary endpoint.

All thirteen variables shown in Table 5 were included in the adjusted model during backward Likelihood ratio (LR) binary logistic regression analysis (cut-off value was 0.5). This revealed the most significant predictors of participants' compliance with referral instruction to the tertiary hospital. Results from the final backward logistic regression model are shown in Table 6. The most important predictor of compliance was health education intervention (OR 4.73; 95% CI 2.87–7.79; p<0.001). Next, compliance rate was positively and significantly affected by participants' self-perception of vision problem (OR 2.02; 95% CI 1.02–4.01; p = 0.045) and participants' income (OR 1.24; 95% CI 1.06–1.44; p = 0.008). The Hosmer and Lemeshow test result was non-significant which indicated that the model is a good fit (p = 0.151) and can significantly predict outcome.

**Table 2. Comparison of primary outcome between intervention and control group.**

| Variables | Intervention N (%) (N = 143) | Control Group N (%) (N = 156) | P-value |
|---|---|---|---|
| Primary Outcome (*Increased referral compliance, i.e. return to tertiary hospital*) | | | |
| Referral compliance | 92 (64.3%) | 44 (28.2%) | <0.001[a] |
| Referral non-compliance | 51 (35.7%) | 112 (71.8%) | |

[a] Pearson chi-square test.

**Table 3. Number of participants who responded positively to knowledge measures (secondary outcome) during pre and post intervention.**

| Knowledge related measures | Study arm | Intervention Group N (%) N = 139 | Control Group N (%) N = 151 |
|---|---|---|---|
| Knows that long-term uncontrolled diabetes might be a cause for a vision problem known as DR | Pre-intervention | 12 (8.6) | 20 (13.2) |
| | Post-intervention | 139 (100) | 59 (39.1) |
| Knows the symptoms of DR | Pre-intervention | 12 (8.6) | 17 (11.3) |
| | Post-intervention | 139 (100) | 28 (18.5) |
| Knows that the onset of DR can be delayed | Pre-intervention | 22 (15.8) | 26 (17.2) |
| | Post-intervention | 131 (94.2) | 21 (13.9) |
| Knows that DR can be treated | Pre-intervention | 111 (79.9) | 136 (90.1) |
| | Post-intervention | 136 (97.8) | 136 (90.1) |
| Correctly indicated one of the symptoms of DR | Pre-intervention | 11 (7.9) | 14 (9.3) |
| | Post-intervention | 134 (96.4) | 50 (33.1) |
| Correctly indicated one of the treatment option for DR | Pre-intervention | 18 (12.9) | 33 (21.9) |
| | Post-intervention | 131 (94.2) | 35 (23.2) |
| Understands impact of non-compliance on vision | Pre-intervention | 132 (95.0) | 128 (84.8) |
| | Post-intervention | 136 (97.8) | 129 (85.4) |
| Knows that a trained Eye Consultant at tertiary hospital provides eye care services | Pre-intervention | 30 (21.6) | 32 (21.2) |
| | Post-intervention | 139 (100) | 68 (45.0) |
| Knows the days and times when Eye Consultant at tertiary hospital provides eye care services | Pre-intervention | 31 (22.3) | 30 (19.9) |
| | Post-intervention | 133 (95.7) | 30 (19.9) |

DR stands for Diabetic Retinopathy.

## Post-hoc sub-group analyses

Although more women than men presented in the baseline survey of this study (shown in S1 Table), more men were likely to visit referred facility (shown in Table 5). Of the total participants, 10.4% (N = 31) experienced physical disability as a result of their existing vision problems ('Strongly agree' and 'Agree' categories taken into account as shown in S1 Table). Of the total study participants, 8% (N = 24) said that they depended on someone's assistance to perform day-to-day tasks as a result of vision problem ('Strongly agree' and 'Agree' categories taken into account as shown in S2 Table). Out of the 24 participants who depended on someone else's assistance, only 37.5% (N = 9) visited the referred facility (as shown in S2 Table). However, it was found that among these compliant participants (N = 9), referral compliance was better in participants who received the intervention (N = 6) compared to those who did not (N = 3).

## Discussion

### Analysis of results

In our study, personalized health education intervention was found to be the most important statistically significant predictor of referral compliance rate. In this study the intervention

**Table 4. Comparison of number of participants who responded negatively to knowledge measures during pre-intervention.**

| Knowledge related measures | Responses (pre and post intervention) | Intervention Group N (%) N = 127 | Control Group N (%) N = 125 | P-value[a] |
|---|---|---|---|---|
| Knows that long-term uncontrolled diabetes might be a cause for a vision problem known as DR | Negative to Positive | 127 (100) | 39 (29.8) | <0.001 |
|  | Negative to Negative | 0 (0) | 92 (70.2) |  |
| Knows the symptoms of DR | Negative to Positive | 127 (100) | 11 (8.2) | <0.001 |
|  | Negative to Negative | 0 (0) | 123 (91.8) |  |
| Knows that the onset of DR can be delayed | Negative to Positive | 109 (97.2) | 0 (0) | <0.001 |
|  | Negative to Negative | 8 (6.8) | 125 (100) |  |
| Knows that DR can be treated | Negative to Positive | 25 (89.3) | 0 (0) | <0.001 |
|  | Negative to Negative | 3 (10.9) | 15 (100) |  |
| Correctly indicated one of the symptoms of DR | Negative to Positive | 123 (96.1) | 36 (26.3) | <0.001 |
|  | Negative to Negative | 5 (3.9) | 101 (73.7) |  |
| Correctly indicated one of the treatment option for DR | Negative to Positive | 113 (93.4) | 2 (1.7) | <0.001 |
|  | Negative to Negative | 8 (6.6) | 116 (98.3) |  |
| Understands impact of non-compliance on vision | Negative to Positive | 5 (71.4) | 1 (4.3) | 0.001 |
|  | Negative to Negative | 2 (28.6) | 22 (95.7) |  |
| Knows that a trained Eye Consultant at tertiary hospital provides eye care services | Negative to Positive | 109 (100) | 36 (30.3) | <0.001 |
|  | Negative to Negative | 0 (0) | 83 (69.7) |  |
| Knows the days and times when Eye Consultant at tertiary hospital provides eye care services | Negative to Positive | 102 (94.4) | 0 (0) | <0.001 |
|  | Negative to Negative | 6 (5.6) | 121 (100) |  |

[a]Pearson chi-square test.

DR stands for Diabetic Retinopathy.

improved knowledge of DR among participants, consistent with other studies that had reported that health education interventions have been successful in increasing screening rates for retinopathy [26–28]. The face-to-face interaction with participants using easy local language and pictorial tools apparently helped to promote understanding among participants. Participants most probably felt empowered to make informed decision to access DFE after receiving specific information about where, when, how and from whom to seek advanced DR management services at a very small cost. This apparently resulted in reduced perceived barriers and increased perceived benefits among participants. When coupled with cues to action (reminder and follow-up), this resulted in the uptake of health services, and this concept is consistent with the Health Belief Model [29].

The second most important predictor of compliance was participant' self-perception of vision problem. Where participants felt that their vision was already affected, there seemed to be a sense of urgency and possibly fear that vision may get worse without timely medical intervention. The perceived susceptibility and severity after the health education as per the Health Belief Model [29], have probably led to increased compliance.

Another important predictor of compliance was participant's income or ability to pay for transportation and logistics required to attend the referred facility (possibly along with an accompanied person). The better financially resourced participants were significantly more likely to visit the referred hospital for further check-up.

## Field experience

Health education, for it to be effective, must be focused, personalized and suitably adapted to local context. It must be appropriate from cultural geographical and social perspectives [30].

**Table 5. Association of primary outcome with other variables.**

| Variables | Compliant N (%) N = 136 | Non-compliant N (%) N = 163 | P-value |
|---|---|---|---|
| **Intervention Group** | | | |
| Health Education Intervention | 92 (67.6) | 51 (31.3) | <0.001[a] |
| Standard Care | 44 (32.4) | 112 (68.7) | |
| **Sex** | | | |
| Women | 62 (45.6) | 92 (56.4) | 0.061[a] |
| Men | 74 (54.4) | 71 (43.6) | |
| **Age Group** | | | |
| ≤29 | 2 (1.5) | 5 (3.1) | 0.476[b] |
| 30–39 | 19 (14.0) | 19 (11.7) | |
| 40–49 | 40 (29.4) | 43 (26.4) | |
| 50–59 | 36 (26.5) | 42 (25.8) | |
| 60+ | 39 (28.7) | 54 (33.1) | |
| **Education** | | | |
| Grade 10 or below | 65 (47.8) | 88 (54.0) | 0.029[b] |
| Passed Grade 10 or has higher qualification | 68 (50.0) | 74 (45.4) | |
| Vocational or others | 3 (2.2) | 1 (0.6) | |
| **Income Range (BDT)** | | | |
| 0–4999 | 71 (52.2) | 97 (59.5) | 0.044[b] |
| 5000–9999 | 7 (5.1) | 11 (6.7) | |
| 10,000–14,999 | 12 (8.8) | 27 (16.6) | |
| 15,000–19,999 | 15 (11.0) | 5 (3.1) | |
| 20,000–49,999 | 26 (19.1) | 23 (14.1) | |
| 50,000 + | 5 (3.7) | 0 (0) | |
| **Travel time in minutes (home to tertiary hospital)** | | | |
| ≤ 60 mins | 68 (50.7) | 71 (43.6) | 0.296[b] |
| 61–90 mins | 23 (16.9) | 30 (18.4) | |
| 91–120 mins | 21 (15.4) | 35 (21.5) | |
| ≥ 121 mins | 23 (16.9) | 27 (16.6) | |
| **Vision problem (self perception)** | | | |
| Yes | 118 (86.8) | 128 (78.5) | 0.063[a] |
| No | 18 (13.2) | 35 (21.5) | |
| **Experiences disability as a result of vision problem** | | | |
| Strongly agree | 2 (1.5) | 3 (1.8) | 0.625[b] |
| Agree | 11 (8.1) | 15 (9.2) | |
| Neutral | 2 (1.5) | 3 (1.8) | |
| Disagree | 121 (89.0) | 142 (87.1) | |
| Strongly disagree | 0 (0.0) | 0 (0.0) | |
| **Requires someone else's assistance to perform day to day activities as a result of vision problem** | | | |
| Strongly agree | 0 (0.0) | 1 (0.6) | 0.282[b] |
| Agree | 9 (6.6) | 14 (8.6) | |
| Neutral | 1 (0.7) | 3 (1.8) | |
| Disagree | 125 (91.9) | 144 (88.3) | |
| Strongly disagree | 1 (0.7) | 1 (0.6) | |
| **Service received at referral point (diabetes hospital)** | | | |

(*Continued*)

**Table 5.** (Continued)

| Variables | Compliant N (%) N = 136 | Non-compliant N (%) N = 163 | P-value |
|---|---|---|---|
| **Waiting time** | | | 0.207[b] |
| <30 minutes | 31 (22.8) | 46 (28.2) | |
| 30–60 minutes | 96 (70.6) | 110 (67.5) | |
| >60 minutes | 9 (6.6) | 7 (4.3) | |
| **Eye screening duration at diabetes hospital** | | | 0.623[b] |
| <20 minutes | 60 (44.1) | 72 (44.2) | |
| 20–40 minutes | 58 (42.6) | 78 (47.9) | |
| >40 minutes | 18 (13.2) | 13 (8.0) | |
| **Counsel time** | | | 0.234[b] |
| <10 minutes | 81 (59.6) | 108 (66.3) | |
| 10–20 minutes | 53 (39.0) | 53 (32.5) | |
| >20 minutes | 2 (1.5) | 2 (1.2) | |
| **Referral** | | | |
| **Overall referral clarity** | | | 0.840[a] |
| Yes | 130 (95.6) | 155 (95.1) | |
| No | 6 (4.4) | 8 (4.9) | |

[a]Pearson chi-square test.

[b]Mann-Whitney U test.

In our experience, participants were found to be very pleased to participate in an interactive personalized learning session, as they felt 'cared-for'.

Telephonic reminder system was the most preferred mode of reminder since all study participants owned or had access to a personal cell phone. Participants were more likely to respond eagerly and provide verbal commitment (by confirming a date) to visit the tertiary hospital during the first telephonic reminder on Day 7 compared with reminders provided on Day 30 and 90. Timely intervention is particularly vital for priority patients such as those suffering from a disability. In our study, the health education resulted in increased DFE among those who faced physical disability as a result of vision problem and required someone else's assistance to perform day-to-day tasks.

In countries with limited resources and high DR prevalence rates, ensuring customized home-based personalized health education even for people with advanced DR maybe costly and challenging. A more cost-effective solution may be to utilize the existing network of CHWs to provide information about DR, and where when and how to access nearby DR management services.

**Table 6. Multivariate binary logistic regression of the predictors of participants' referral compliance.**

| Variable | B | OR | Results at final step 95% CI for OR Lower Upper | | P-value | Nagelkerke R² |
|---|---|---|---|---|---|---|
| Intervention type | 1.553 | 4.73 | 2.89 | 7.79 | <0.001 | 0.21* |
| Self-perception (vision problem) | 0.703 | 2.02 | 1.02 | 4.01 | 0.045 | |
| Monthly income | 0.211 | 1.24 | 1.06 | 1.44 | 0.008 | |

Hosmer and Lemeshow Test result indicates that model is non-significant (p = 0.151).

Prediction percentage correct for above model = 68.2%.

*The Nagelkerke $R^2$ is 0.21 for this model.

## Similarity with other global studies

Our findings are consistent with the findings of similar studies trialed in developed countries [17, 18] where increase in DFE was reported as a result of health education intervention. Our multicomponent and locally adapted intensive health education intervention succeeded to attain a referral compliance rate that was 36.1% higher (in other words, this is equivalent to 128% increase in interventions group compared to control group). Several RCTs based in developed countries that used varied health education modalities to increase DFE reported a lower compliance rate [21, 31–37].

Apart from the fact that a multicomponent intervention was used in our study, there may be other reasons why it was possible to attain higher compliance rate compared to other global studies. It may be easier to achieve a higher change in intervention group where baseline referral compliance rates are relatively low [17] such as in countries with limited health resources. On the other hand, despite lower education level among participants, and social and infrastructural barriers to accessing health care generally expected in limited resource settings, the compliance rate among intervention group was more than what was initially expected during study design.

## Future implications

To sustain increased DFEs among persons with T2DM, personalized multicomponent behavior change strategies used in this trial may be tested and provided at the institutions where DR management services are either available within the hospital or in the community. Additional demonstrative strategies such as using audio-visual modalities within hospital premises may be explored and adopted. Further studies may be conducted in LMICs to understand social infrastructural and other systemic barriers to accessing DR management services, so as to devise and trial other effective interventions to increased referral compliance.

## Study limitations

This study may not be generalized for the entire population who are suffering from diabetes. Participants of this study were registered with a diabetes hospital and therefore generally aware about diabetes mellitus.

Furthermore, in this study, the referred facility was located in the same district as participant's residence as well as the diabetes hospital. In areas where participants live further away, or where referred facility is further away, the referral compliance rate may not improve significantly among those who have been provided with health education.

## Supporting information

**S1 Table. Baseline characteristics of study participants according to subgroups.**
(DOCX)

**S2 Table. Effect of intervention on referral compliance for patients who require someone else's assistance.**
(DOCX)

**S1 File. Informed consent form in English.**
(PDF)

**S2 File. Informed consent form in Bengali.**
(PDF)

**S3 File. Questionnaire in English.**
(PDF)

**S4 File. Questionnaire in Bengali.**
(PDF)

**S5 File. Health education content.**
(DOCX)

**S6 File. CONSORT checklist.**
(DOC)

**S7 File. Trial protocol.**
(PDF)

**S8 File. Approved protocol of ethics committee.**
(PDF)

## Acknowledgments

The authors are grateful for the support provided during the study by Dr. Yoshiaki Kiuchi, Professor, Department of Ophthalmology and Visual Sciences, Hiroshima University, Japan, Dr. Md. Shafiqul Islam, Assistant Professor, Department of Ophthalmology, Sher-e-Bangla Medical College & Hospital, Barishal, Bangladesh, and Dr. Md. Ataur Rahman (Chunnu), Senior Eye Consultant, Advocate Hemayet Uddin Ahmed Diabetic & General Hospital, Barishal, Bangladesh. Community health workers have been tremendously supportive in visiting participants' homes given the challenging climate and transportation infrastructure in hard-to-reach areas in Barishal district.

## Author Contributions

**Conceptualization:** Zara Khair, Md Moshiur Rahman, Michiko Moriyama.

**Data curation:** Zara Khair.

**Formal analysis:** Zara Khair, Kana Kazawa, Abu S. G. Faruque.

**Investigation:** Zara Khair.

**Methodology:** Zara Khair, Md Moshiur Rahman, Mohammod Jobayer Chisti, Michiko Moriyama.

**Project administration:** Zara Khair, Abu S. G. Faruque, Michiko Moriyama.

**Resources:** Zara Khair, Md Moshiur Rahman, Abu S. G. Faruque, Mohammod Jobayer Chisti, Michiko Moriyama.

**Supervision:** Zara Khair, Md Moshiur Rahman, Michiko Moriyama.

**Validation:** Zara Khair, Mohammod Jobayer Chisti.

**Visualization:** Zara Khair, Yasmin Jahan.

**Writing – original draft:** Zara Khair.

**Writing – review & editing:** Md Moshiur Rahman, Kana Kazawa, Yasmin Jahan, Abu S. G. Faruque, Mohammod Jobayer Chisti, Michiko Moriyama.

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
