## [Decision Letter · Decision Letter 0]

20 Aug 2020

PONE-D-20-18885

Health education improves referral compliance of persons with probable Diabetic Retinopathy: A randomized controlled trial

PLOS ONE

Dear Dr. Khair,

Thank you for submitting your manuscript to PLOS ONE. After careful consideration, we feel that it has merit but does not fully meet PLOS ONE’s publication criteria as it currently stands. Therefore, we invite you to submit a revised version of the manuscript that addresses the points raised during the review process.

We look forward to receiving your revised manuscript.

Kind regards,

Antonio Palazón-Bru, PhD

Academic Editor

PLOS ONE

Journal Requirements:

Reviewers' comments:

Reviewer's Responses to Questions

**Comments to the Author**

1. Is the manuscript technically sound, and do the data support the conclusions?

Reviewer #1: Yes

Reviewer #2: Yes

Reviewer #3: Partly

2. Has the statistical analysis been performed appropriately and rigorously? 

Reviewer #1: Yes

Reviewer #2: Yes

Reviewer #3: No

3. Have the authors made all data underlying the findings in their manuscript fully available?

Reviewer #1: Yes

Reviewer #2: No

Reviewer #3: Yes

4. Is the manuscript presented in an intelligible fashion and written in standard English?

Reviewer #1: Yes

Reviewer #2: No

Reviewer #3: Yes

5. Review Comments to the Author

Reviewer #1: This is an interesting article, with findings that could impact care of patients with diabetic retinopathy in developing countries. Overall, the manuscript is too long, and much of the introduction and discussion can likely be shortened. There are other issues to consider:

1. The introduction is very long. It can likely be cut significantly. There is considerable mention of previous studies that are very similar in scope and purpose, which makes the reader question why this study is necessary. The authors eventually state that this study is unique in that it is conducted in a low/middle income country, but that statement is not made until line 130. It should be made earlier in the text.

2. Line 70--the authors state that there is no treatment to restore vision that has already been lost, but that is not technically true. Anti-VEGF medications can improve vision in the case of DME, though it may be true that those treatments are not readily available in certain communities.

3. Lines 114-118 refer to a "study" that observed a large number of patients who were being referred to for dilated exams were non-compliant, though there is no reference provided. Does that mean you are referring to the current study? If so, then this is not background for the paper (or hypothesis driving). It would, however, be interesting and relevant to know what baseline compliance rates were prior to performing the intervention/conducting the study.

4. It appears that as long as the patients presented for their fundus exam within the study time frame, they were considered compliant, which means that patients who presented up to a year after referral were still considered compliant. There are many scenarios in which that would not really be considered "compliance" given the significant progression of disease that may occur in that time frame, so it would be important to note what the average time between referral to presentation/appointment scheduling was.

5. Table 1 and 2 are included in the methods section, but would be more appropriate in a results section.

6. Re: Table 3; It is interesting that there was no difference between the education and the control group on the understanding of the impact of non-compliance on vision, or the understanding that DR can be treated. Why, then, was compliance improved when there was no difference in the patients' risk perception? This is something that should be further discussed. What part of the education do the authors think was most impactful? Many would assume that a better understanding of potential risk of non-compliance and potential benefit of compliance would encourage patients to be more compliant, however the data demonstrates no difference in understanding between those in the control vs treatment group.

7. The association of eye screening duration with referral compliance is interesting. Is it possible that just spending longer with the patient fostered a sense of urgency in the patient? This should be considered in the manuscript and for further research.

8. The authors should consider a survey of the patients who were compliant to better understand the factors that encouraged them to present.

9. Lines 482-488: The authors discuss cost of scaling a program such as what they have presented. It would be interesting to consider the difference in the cost of scaling a program like that presented in the paper to the cost, for example, of increasing access to care (via programs like eye camps, etc.)

10. Lines 493-505 and line 531-532: It is not really accurate to say that the program presented in the study is "more successful" than previously published programs, as the two are not really comparable. Previous studies, were not only conducted in higher income countries, but also measured rates of DR screening, whereas this study refers to rates of presentation for further care after a positive screening exam. These are two different cohorts of patients, with very different risks, so should not be compared in a head-to-head fashion.

Reviewer #2: The study includes only a modest number of subject compared to other similar studies.

The Health Education is too cumbersome and time-consuming to be useful on a large scale. It involves the 30-40 initial interview. The drive to the patient's house, the flip charts, the brochure, the 15 minute phone calls, the videotape etc. The patients also often had a long drive to the tertiary hospital. Even with all this work, the follow-up rate was still only 67%. The point is, wouldn't it be easier to screen the retina for DR at the initial visit? Have a doctor present t or take a photo with an inexpensive camera for telemedicine. It seems like it would save so much time and effort for both the health care team and patient.

It's not quiet clear how the compliance rate was measured? Was it a visit to the tertiary hospital in the 6 months following the initial visit. It's also not quite clear about how the knowledge test was administered.

Reviewer #3: The manuscript entitled ‘Health education improves referral compliance of persons with probable Diabetic Retinopathy: A randomized controlled trial’ with the aim to evaluate the efficacy of a culturally, geographically and socially appropriate, locally adapted five month-long health education on referral compliance of participants.

The manuscript requires improvement based on the following comments.

Comments

2 decimal points are sufficient for OR and 95%CI figures.

Methods

Trial design

Page 7, ethics committee approval to be stated in the method section.

Sample size calculation

Page 10, 1 or 2 tailed test to be stated.

Statistical methods

Page 13 Line 291, for Independent t tests and chi-squared tests, s to be omitted.

Page 13 Line 294, proper citation including publisher name for SPSS to be stated.

Results

Page 14 Table 1, for p value readings, actual p value to be given.

However, based on the CONSORT guidelines, all statistical tests for group comparison at baseline to be avoided. Symbol <= to be replaced with symbol ≤. n to be provided apart from percentage figures. N(%) to be placed on the first row. Symbol % before the variable name to be removed.

Page 14 Table 2, the figures for non-compliance to be displayed.

Page 14 Table 1 & 2, total N to be provided.

Page 14 & 15, Table 1, 2, 3 to be placed in the results section.

Page 15, what type of chi-square test used here? Chi-square test of independence is not suitable for repeated measures.

Page 17 Line 365 typo Reaults.

Page 18 Table 4, N to be stated on top. The words ‘to 1 Decimal place)’ to be removed. Symbol % for individual figures to be omitted. Likewise with Table S6.

Page 18 Table 4, the presentation to include figures for ‘No’ apart from ‘Yes’. If not ‘n’ for each subcategory for each variable to be provided. Symbol <=, >= to be replaced with ≤, ≥ respectively. Total N to be stated. Symbol % for individual figures to be omitted. The selection criteria for variable(s) selection in the univariate analysis for the inclusion in the adjusted model to be clearly stated. If based on the referral compliance rate improvement, what was the figures chosen or if based on the p value, what was the cut off. The name Pearson chi-square test or chi-squared test to be standardized where appropriate.

Page 19 Line 378-379, the sentence ‘multivariate binary backward logistic regression modeling.’ to be revised. The exact type of backward elimination method to be stated.

Page 19 Line 382-387, p value to be placed after 95%CI.

Page 20, Annex 2?

Page 20 Line 412, Table 4 to be written as Table S7.

Page 20 Table 5, title to be revised. The word ‘Backward’ to be omitted. Exp B to be replaced with OR.

6. PLOS authors have the option to publish the peer review history of their article (what does this mean?). If published, this will include your full peer review and any attached files.

Reviewer #1: No

Reviewer #2: No

Reviewer #3: No

---

## [Author Response · Author response to Decision Letter 0]

27 Sep 2020

Date: 25th September, 2020

To

Antonio Palazón-Bru, PhD 

Academic Editor 

PLOS ONE 

Dear Dr. Palazón-Bru, 

Greetings! 

We thank you and the all the respected Reviewers for thoroughly assessing and providing constructive feedback on our manuscript titled ‘Health education improves referral compliance of persons with probable Diabetic Retinopathy: A randomized controlled trial’ [PONE-D-20-18885]. 

We have addressed all comments in the following pages and are pleased to resubmit our revised manuscript for your consideration for publication in PLOS ONE. We believe that the paper has improved significantly in quality, relevance and presentation, since all Reviewers’ comments have been addressed.

On behalf of my co-authors, I thank you for your consideration of this resubmission. We appreciate your highly effective engagement and look forward to your positive response. 

Sincerely 

Zara Khair 

(on behalf of co-authors)

Response to Reviewer # 1

This is an interesting article, with findings that could impact care of patients with diabetic retinopathy in developing countries. Overall, the manuscript is too long, and much of the introduction and discussion can likely be shortened. 

Thank you for your valuable comments. We have revised the article accordingly, and have now shortened the introduction and discussion sections significantly. 

There are other issues to consider: 

1. The introduction is very long. It can likely be cut significantly. There is considerable mention of previous studies that are very similar in scope and purpose, which makes the reader question why this study is necessary. The authors eventually state that this study is unique in that it is conducted in a low/middle income country, but that statement is not made until line 130. It should be made earlier in the text. 

We agree with Reviewer’s comments and have reduced the Introduction section as well as the Discussion section significantly. 

The fact that this study is unique in that it is conducted in a low/middle income country is now mentioned much earlier in the Introduction section (page 5, lines 87 – 91).

2. Line 70--the authors state that there is no treatment to restore vision that has already been lost, but that is not technically true. Anti-VEGF medications can improve vision in the case of DME, though it may be true that those treatments are not readily available in certain communities.

We agree with Reviewer’s comment and have omitted the sentence that stated that ‘there is no treatment to restore vision that has already been lost’. 

3. Lines 114-118 refer to a "study" that observed a large number of patients who were being referred to for dilated exams were non-compliant, though there is no reference provided. Does that mean you are referring to the current study? If so, then this is not background for the paper (or hypothesis driving). It would, however, be interesting and relevant to know what baseline compliance rates were prior to performing the intervention/conducting the study.

That is correct - in our first manuscript submission, in lines 114 – 118, ‘study’ referred to our current study. During first submission, we included this information as we wanted readers to understand that similar high baseline non-compliance rates reported in global studies was also found in our current study. To avoid confusion, we have now removed that sentence. 

4. It appears that as long as the patients presented for their fundus exam within the study time frame, they were considered compliant, which means that patients who presented up to a year after referral were still considered compliant. There are many scenarios in which that would not really be considered "compliance" given the significant progression of disease that may occur in that time frame, so it would be important to note what the average time between referral to presentation/appointment scheduling was.

We agree with this suggestion, and have now included specific information regarding ‘time between referral to presentation’ in page 6, line 116 and in page 10, lines 205-208.

5. Table 1 and 2 are included in the methods section, but would be more appropriate in a results section. 

We have now included Table 1 and 2 in the Results section (page 14-15)

6. Re: Table 3; It is interesting that there was no difference between the education and the control group on the understanding of the impact of non-compliance on vision, or the understanding that DR can be treated. Why, then, was compliance improved when there was no difference in the patients' risk perception? This is something that should be further discussed. What part of the education do the authors think was most impactful? Many would assume that a better understanding of potential risk of non-compliance and potential benefit of compliance would encourage patients to be more compliant, however the data demonstrates no difference in understanding between those in the control vs treatment group. 

We thank Reviewer for this highly relevant and thought-provoking questions and suggestions. We have now provided our further insights and explanations in the Statistical Methods and Discussion section. 

Please note that as per suggestion made by Reviewer#3, we have revised the statistical analysis for the secondary outcome (‘increase in knowledge’). 

In the Statistical Analysis section (page 13, Lines 276-284), we have now mentioned that chi-square test was performed for each knowledge related question. For the secondary outcome, we compared the proportion of participants who provided correct (or positive) responses during pre and post intervention surveys. This comparison was done between the two study arms. In addition, chi-square test was performed for each knowledge related question. There were no participants who initially provided correct (or positive) response at baseline and later incorrect (or negative) response during the post intervention survey. Therefore, we conducted chi-square test only on those participants who initially provided incorrect (or negative) responses. 

Our revised Tables 3 and 4 (pages 16-17) now highlight that there have been significant differences across all of the knowledge measures. We have updated Discussion section accordingly.

Please note that during our first manuscript submission, we did not exclude 9 drop out participants from the Pre-intervention survey result. Now, we have excluded the drop out participants from pre and post intervention survey results, and compared using chi-square test. Therefore, our results have also been updated accordingly in Table 3 and 4 (pages 16-17).

7. The association of eye screening duration with referral compliance is interesting. Is it possible that just spending longer with the patient fostered a sense of urgency in the patient? This should be considered in the manuscript and for further research. 

We thank Reviewer for this valuable comment. 

Please note that as per suggestion made by Reviewer#3, we have now conducted regression analysis with a cut-off value of 0.5 and included all 13 variables used to conduct univariate analysis shown Table 5 (page 18-19). As a result, the final regression model does not have the ‘eye screening duration’ as one of the three significant predictors. Please refer to Table 6 in page 20. All other predictors (‘heath education intervention’ and ‘income’) reported during first submission of manuscript were still found to be significant. 

8. The authors should consider a survey of the patients who were compliant to better understand the factors that encouraged them to present. 

We agree with this suggestion. Reviewer will be pleased to know that authors are currently considering a study on complaint participants to better understand the factors that encouraged them to present.

9. Lines 482-488: The authors discuss cost of scaling a program such as what they have presented. It would be interesting to consider the difference in the cost of scaling a program like that presented in the paper to the cost, for example, of increasing access to care (via programs like eye camps, etc.) 

We have now indicated that a more cost-effective approach would be to utilize the existing network of Community Health Workers to disseminate information about DR and nearby available service (page 22, lines 446-448).

10. Lines 493-505 and line 531-532: It is not really accurate to say that the program presented in the study is "more successful" than previously published programs, as the two are not really comparable. Previous studies, were not only conducted in higher income countries, but also measured rates of DR screening, whereas this study refers to rates of presentation for further care after a positive screening exam. These are two different cohorts of patients, with very different risks, so should not be compared in a head-to-head fashion. 

Authors agree with this review. All references to current study being ‘more successful’ have now been omitted. Statements where direct comparisons were made have been modified (Page 23, lines 451 – 457).

Response to Reviewer # 2

1. The study includes only a modest number of subject compared to other similar studies. 

We thank Reviewer for this insight. We would humbly like to state that to conduct a high quality Randomized Controlled Trial, we used the superiority sample calculation formula to ensure a sufficient power of 90% with 95% CI, so that study results are considered significant (page 8, lines 154 – 167). 

The sample size of our trial (N=299) is more than some of the other similar published studies as well, e.g. sample size used in similar published RCTs was reported to be 132 by Anderson et al. [21] and 280 by Basch et al [37].

2. The Health Education is too cumbersome and time-consuming to be useful on a large scale. It involves the 30-40 initial interview. The drive to the patient's house, the flip charts, the brochure, the 15 minute phone calls, the videotape etc. The patients also often had a long drive to the tertiary hospital. Even with all this work, the follow-up rate was still only 67%. The point is, wouldn't it be easier to screen the retina for DR at the initial visit? Have a doctor present t or take a photo with an inexpensive camera for telemedicine. It seems like it would save so much time and effort for both the health care team and patient. 

We thank the Reviewer for discussing valuable insights on a very relevant area. 

All participants in our study had already undergone a preliminary DR screening at diabetes hospital using low-resolution fundus camera (please kindly refer to page 7, line 141). 

In our study, we were required to implement a multi-component health education intervention for several reasons. Not only was the education level of community generally low in our research setting, but the health system is also quite resource poor (e.g. lack of ophthalmic health personnel, low referral compliance, etc.). Moreover, the psychological aspect of our study participants had to be taken into account – persons with diabetes often forego their appointments even in developed countries because of their mental health status among other reasons. 

We have now replaced with the phrase ‘behavior change strategies used in this trial may be tested and provided at the institutions’ in page 23, line 470-471. We have updated the ‘Future Implications’ section in page 23-24 as well. Institutionalization of health education intervention would mean that those who are detected with probable DR at diabetes hospitals might be right away provided with health education interventions to motivate them to undergo DFE. This would eventually save a lot of time and effort for health care team and persons with diabetes. 

3. It's not quiet clear how the compliance rate was measured? Was it a visit to the tertiary hospital in the 6 months following the initial visit. It's also not quite clear about how the knowledge test was administered. 

We thank the Reviewer for specifying this information gap in our manuscript. We have now included information about how compliance rate was measured (page 10, lines 205–208). We have also detailed how knowledge test was administered (page 10, lines 214-216). 

Response to Reviewer # 3

1. Comments: 2 decimal points are sufficient for OR and 95%CI figures. 

We have rounded OR and 95% CI figures to 2 decimal points.

2. Methods: Trial design Page 7, ethics committee approval to be stated in the method section. 

We have now mentioned about ethic committee approval in the methods section (page 6, line 117-118).

3. Sample size calculation: Page 10, 1 or 2 tailed test to be stated. 

We have now mentioned about 2-tailed test (Page 9, line 179).

4. Statistical methods: Page 13 Line 291, for Independent t tests and chi-squared tests, s to be omitted.

We have omitted ‘s’ from ‘tests’ (page 12, line 266).

5. Page 13 Line 294, proper citation including publisher name for SPSS to be stated. 

We have now mentioned proper citation including publisher name for SPSS (page 13, line 287).

6. Results: Page 14 Table 1, for p value readings, actual p value to be given. However, based on the CONSORT guidelines, all statistical tests for group comparison at baseline to be avoided. Symbol <= to be replaced with symbol ≤. n to be provided apart from percentage figures. N(%) to be placed on the first row. Symbol % before the variable name to be removed. 

We thank the Reviewer for this important insight and suggestion. We have removed the ‘p value’ column from Table 1 (page 14-15). We have replaced the symbol <= with symbol ≤. We have provided ‘n’ apart from percentage figures, placed N (%) on the first row, and removed the symbol % before the variable names. 

7. Page 14 Table 2, the figures for non-compliance to be displayed. Page 14 Table 1 & 2, total N to be provided. 

We have now displayed the figures for non-compliance in Table 2 (page 15). We have also provided total N in Table 1 and Table 2 (page 14-15). 

8. Page 14 & 15, Table 1, 2, 3 to be placed in the results section. 

Tables 1, 2 and 3 are now placed in the Results section. 

9. Page 15, what type of chi-square test used here? Chi-square test of independence is not suitable for repeated measures. 

We thank Reviewer for this important observation. In the Statistical Analysis section (page 13, line 276-284), we have now mentioned that chi-square test was performed for each knowledge related question (line 278-279). 

For the secondary outcome, we compared the proportion of participants who provided correct (or positive) responses during pre and post intervention surveys. This comparison was done between the two study arms. In addition, chi-square test was performed for each knowledge related question. There were no participants who initially provided correct (or positive) response at baseline and later incorrect (or negative) response during the post intervention survey. Therefore, we conducted chi-square test only on those participants who initially provided incorrect (or negative) responses. 

Please note that during our first manuscript submission, we did not exclude 9 drop out participants from the Pre-intervention survey result. Now, we have excluded the drop out participants from pre and post intervention survey results, and compared using chi-square test. Therefore, our results have also been updated accordingly (Table 3 and 4). 

10. Page 17 Line 365 typo Reaults. 

We have corrected this typo (Page 18, Line 356).

11. Page 18 Table 4, N to be stated on top. The words ‘to 1 Decimal place)’ to be removed. Symbol % for individual figures to be omitted. Likewise with Table S6. 

In the Table, we have now stated ‘N’ on top, removed the words ‘to 1 Decimal place’, omitted the symbol % for individual figures (previously Table 4, now Table 5 in Page 18). We have done the same for S6 Table and resubmitted S6 Table along with revised manuscript. 

12. Page 18 Table 4, the presentation to include figures for ‘No’ apart from ‘Yes’. If not ‘n’ for each subcategory for each variable to be provided. Symbol <=, >= to be replaced with ≤, ≥ respectively. Total N to be stated. Symbol % for individual figures to be omitted. The selection criteria for variable(s) selection in the univariate analysis for the inclusion in the adjusted model to be clearly stated. If based on the referral compliance rate improvement, what was the figures chosen or if based on the p value, what was the cut off. The name Pearson chi-square test or chi- squared test to be standardized where appropriate. 

For Table 5 (previously Table 4), there are no figures for ‘No’ as it does not apply for the variables presented (we have now mentioned total N in the Table).

We have replaced symbols <= and >= with ≤ and ≥ respectively. We have stated total N, omitted % symbol for individual figures. 

All the thirteen variables shown in Table 5 were included in the adjusted model during logistic regression analysis (cut-off value was 0.5). This has been clearly mentioned now (Page 19, Line 367-369). 

We have now revised the data type of few of the thirteen variables from ‘nominal’ to ‘ordinal’, where appropriate, and this had led to slight changes in the result of the regression as shown in Table 6 (page 20). 

We have now reworded as ‘chi square test’ throughout the manuscript. 

13. Page 19 Line 378-379, the sentence ‘multivariate binary backward logistic regression modeling.’ to be revised. The exact type of backward elimination method to be stated. 

We have now revised the phrase as ‘multivariate analysis with backward Likelihood Ratio (LR) binary logistic regression modeling’.

The exact type of backward elimination method was ‘Likelihood Ratio’, which has now been specified. (Page 19, Line 368 and Page 12, Line 269-270).

14. Page 19 Line 382-387, p value to be placed after 95%CI. 

All p-values are now placed after 95%CI.

15. Page 20, Annex 2? 

We have replaced the word ‘Annex 2’ with ‘S6 Table’. (Page 20, Line 392).

16. Page 20 Line 412, Table 4 to be written as Table S7. 

We have replaced the word ‘Table 4’ with ‘S7 Table’ (Page 20, Line 395).

(We have followed PLOS ONE numbering and naming guideline for supplementary tables.) 

17. Page 20 Table 5, title to be revised. The word ‘Backward’ to be omitted. Exp B to be replaced with OR. 

We have omitted the word ‘Backward’ from the Regression Table title, and replaced ‘Exp B' with ‘OR’ (page 20, now Table 6).

---

## [Decision Letter · Decision Letter 1]

20 Oct 2020

PONE-D-20-18885R1

Health education improves referral compliance of persons with probable Diabetic Retinopathy: A randomized controlled trial

PLOS ONE

Dear Dr. Khair,

Thank you for submitting your manuscript to PLOS ONE. After careful consideration, we feel that it has merit but does not fully meet PLOS ONE’s publication criteria as it currently stands. Therefore, we invite you to submit a revised version of the manuscript that addresses the points raised during the review process.

We look forward to receiving your revised manuscript.

Kind regards,

Antonio Palazón-Bru, PhD

Academic Editor

PLOS ONE

Reviewers' comments:

Reviewer's Responses to Questions

**Comments to the Author**

1. If the authors have adequately addressed your comments raised in a previous round of review and you feel that this manuscript is now acceptable for publication, you may indicate that here to bypass the “Comments to the Author” section, enter your conflict of interest statement in the “Confidential to Editor” section, and submit your "Accept" recommendation.

Reviewer #2: All comments have been addressed

Reviewer #3: (No Response)

2. Is the manuscript technically sound, and do the data support the conclusions?

Reviewer #2: Yes

Reviewer #3: Partly

3. Has the statistical analysis been performed appropriately and rigorously? 

Reviewer #2: Yes

Reviewer #3: (No Response)

4. Have the authors made all data underlying the findings in their manuscript fully available?

Reviewer #2: Yes

Reviewer #3: Yes

5. Is the manuscript presented in an intelligible fashion and written in standard English?

Reviewer #2: Yes

Reviewer #3: Yes

6. Review Comments to the Author

Reviewer #2: (No Response)

Reviewer #3: Minor comments

Table 6, for 95% CI for exp B, exp B to be replaced with OR.

Line 377, it was stated p=0.330 but Line 383 stated p=0.151

7. PLOS authors have the option to publish the peer review history of their article (what does this mean?). If published, this will include your full peer review and any attached files.

Reviewer #2: No

Reviewer #3: No

---

## [Author Response · Author response to Decision Letter 1]

20 Oct 2020

Response to Reviewer # 3

1. Table 6, for 95% CI for exp B, exp B to be replaced with OR.

We have now replaced the term ‘Exp (B)’ with OR. 

2. Line 377, it was stated p=0.330 but Line 383 stated p=0.151

We have fixed this typo in line 377 to read p=0.151.

Therefore both the p-values in Line 377 and Line 383 now read the same.

---

## [Decision Letter · Decision Letter 2]

27 Oct 2020

Health education improves referral compliance of persons with probable Diabetic Retinopathy: A randomized controlled trial

PONE-D-20-18885R2

Dear Dr. Khair,

We’re pleased to inform you that your manuscript has been judged scientifically suitable for publication and will be formally accepted for publication once it meets all outstanding technical requirements.

Kind regards,

Antonio Palazón-Bru, PhD

Academic Editor

PLOS ONE

Additional Editor Comments (optional):

Reviewers' comments:

Reviewer's Responses to Questions

**Comments to the Author**

1. If the authors have adequately addressed your comments raised in a previous round of review and you feel that this manuscript is now acceptable for publication, you may indicate that here to bypass the “Comments to the Author” section, enter your conflict of interest statement in the “Confidential to Editor” section, and submit your "Accept" recommendation.

Reviewer #3: All comments have been addressed

2. Is the manuscript technically sound, and do the data support the conclusions?

Reviewer #3: (No Response)

3. Has the statistical analysis been performed appropriately and rigorously? 

Reviewer #3: (No Response)

4. Have the authors made all data underlying the findings in their manuscript fully available?

Reviewer #3: (No Response)

5. Is the manuscript presented in an intelligible fashion and written in standard English?

Reviewer #3: (No Response)

6. Review Comments to the Author

Reviewer #3: (No Response)

7. PLOS authors have the option to publish the peer review history of their article (what does this mean?). If published, this will include your full peer review and any attached files.

Reviewer #3: No

---

## [Editor Report · Acceptance letter]

4 Nov 2020

PONE-D-20-18885R2 

Health education improves referral compliance of persons with probable Diabetic Retinopathy: A randomized controlled trial 

Dear Dr. Khair:

I'm pleased to inform you that your manuscript has been deemed suitable for publication in PLOS ONE. Congratulations! Your manuscript is now with our production department. 

Kind regards, 

on behalf of

Dr. Antonio Palazón-Bru 

Academic Editor

PLOS ONE